# Classification of Engine Type of Vehicle Based on Audio Signal as a Source of Identification

**Mateusz Materlak [1] and Ewelina Majda-Zdancewicz [2],***

1. 3st Military Centre of Metrology, Military University of Technology, 56-400 Oleśnica, Poland
2. Faculty of Electronics, Military University of Technology, 00-908 Warsaw, Poland
* Correspondence: ewelina.majda@wat.edu.pl

**Abstract:** In this work, a combination of signal processing and machine learning techniques is applied for petrol and diesel engine identification based on engine sound. The research utilized real recordings acquired in car dealerships within Poland. The sound database recorded by the authors contains 80 various audio signals, equally divided. The study was conducted using feature engineering techniques based on frequency analysis for the generation of sound signal features. The discriminatory ability of feature vectors was evaluated using different machine learning techniques. In order to test the robustness of the proposed solution, the authors executed a number of system experimental tests, including different work conditions for the proposed system. The results show that the proposed approach produces a good accuracy at a level of 91.7%. The proposed system can support intelligent transportation systems through employing a sound signal as a medium carrying information on the type of car moving along a road. Such solutions can be implemented in the so-called 'clean transport zones', where only petrol-powered vehicles can freely move. Another potential application is to prevent misfuelling diesel to a petrol engine or petrol to a diesel engine. This kind of system can be implemented in petrol stations to recognize the vehicle based on the sound of the engine.

**Keywords:** sound processing; engine sound; feature extraction; vehicle classification; intelligent transport system





## 1. Introduction

Sound is one of the primary forms of sensory information that we use to perceive our surroundings. The classification of sounds is widely used in several different fields. Because of this, the classification of sounds has become a very popular topic. Fields of application include, for example: multimedia retrieval [1,2], technology medical problems s [3,4], speech recognition [5], speaker recognition [6,7], urban sound classification [8,9], environmental sound classification [10,11], speech emotion recognition [12], animal sound classification [13,14], detection of mechanical failure [15], and many others. In recognition tasks, the basic issue is what to recognize, in other words, what the inputs of the system are.

Sound classification is also employed within intelligent transport systems for analyzing and managing road traffic. This application is extremely well developed and driven by an increase in general traffic volume, primarily in urban agglomerations. The concept of intelligent transportation systems (ITS) refers to the use of unconventional process and organizational transport solutions aimed at supporting the operation of road infrastructure and improving road user safety [16]. The decision-making process associated with these systems is based on analyzing data collected through various sensors [17]. An acoustic signal is one such example [18]. Currently, sound signals in the road traffic space are monitored mainly for the purposes of controlling noise levels. Another application may be the detection of dangerous incidents, such as gunshots, explosions, accidents, collisions, or other distress requiring help [19,20]. Recognition based on acoustic information is possible

if the sound generated by tracking object includes specific features that allow to distinguish it from signals produced by other vehicles [21,22].

One application of ITS technology lies in the enforcement of clean transport zones. In these, only petrol-powered vehicles can freely move [23]. Nowadays, vehicle verification is based on a system of cameras that recognize license plates and use them to assess whether a vehicle has the right to enter the zone [24]. This technology has been already introduced in many European cities [25]. An alternative method is analyzing sound and quickly notifying an unauthorized driver about the prohibition on entering a given area. This is particularly useful for non-local drivers that are not usually familiar with the legislation applicable in a given city.

The article presents a method for vehicle class identification based on recorded sound signals. In this case, vehicle class is construed as the type of engine (petrol or diesel) that the vehicle is fitted with. Internal combustion engines can be divided by different criteria. However, this study is focused on classifying them based on the ignition method, and, hence, the type of consumed fuel. The objective of the experiments conducted by the authors in this paper is to develop a target system that enables petrol and diesel engine identification based on the engine sound by means of digital signal processing, including the use of machine learning algorithms. Such a system could be treated as a part of the application of industrial revolution 4.0 in the ITS sector.

First, the paper presents related works for automatic identification of the engine type. Herein, the authors point out that the general motivation behind developing petrol and diesel engine identification via sound is that, today, there are only a few available solutions to this problem. Currently, engine sounds are mostly used for identifying the type of car (i.e., car, bus, truck, motorcycle, military vehicle) [26–28]. The next section of the on-going work describes our application of feature engineering practices, which means finding sets of parameters to be used as a base to generate feature vectors for modelling the engine sound. In the work, our study utilized the sound database that the authors collected which contains 80 various audio signals, equally divided into diesel and petrol engines. The presented algorithm was then applied to evaluate the signal using a spectrum. Vectors of selected voice descriptors were used in the classification scheme based on different neural networks. The subsequent section describes a test of the robustness of the proposed solution. In undertaking this, the authors executed a number of system experimental tests, including different work conditions for the proposed system. Finally, the last section summarizes the research, compares our results with other research, and points to the direction of further research.

## 2. Related Works

Automatic identification of engine type is a research area that is not widely analyzed in the world literature. Early publications regarding automated acoustic vehicle recognition algorithms were focused mainly on military vehicle signals [29], in order to develop a system that improves surveillance for security. As part of the experiments, analysis methods such as FFT (fast Fourier transform) [21], STFT (short time Fourier transform) [26], time-varying autoregressive (TVAR) combined with low-order discrete cosine transform (DCT) [30], MFCC (Mel-frequency cepstral coefficients) [31], and wavelet packet were used [32].

Current studies focused on the use of the acoustic signal generated by the vehicle engine have been presented in relation to the identification of the type of machine. The authors of these have proposed utilizing so-called "machine biometrics", which is understood as the identification of the vehicle brand. Based on the completed research, 22 sound features were extracted and their discrimination capabilities were tested in combination with nine different machine learning classifiers, towards identifying five vehicle manufacturers. The experimental results revealed the ability of the proposed biometrics to identify vehicles with high accuracy up to 98% for the case of the multilayer perceptron (MLP) neural network model [27]. Generally, such research usually focuses on recognizing classes for various vehicles, but not for engine type based on sounds [31,33].

Using audio signal as the basis for identifying the type of engine, was discussed in the article [34]. As part of the research, the authors defined the characteristics of the signal based on the FFT transform, and used the SVM network. However, they employed a less numerous acoustic database. Another interesting research focuses on detecting V6 and V8 engines based on audio signal [35].

Most of the studies presented in the literature describe the identification of the type of machine defined, e.g., a car, bus, truck, or motorcycle [36–38]. The research proposed by the authors in the current study is focused on type of engine (petrol or diesel), regardless of type of machine. The solutions proposed in this area are very limited. First of all, the authors of works in the literature have used different databases and limited numbers of recordings with a limited diversity of recorded signals [34,35]. Greater diversity corresponds to creating a common database utilizing the products of different vehicle manufacturers and different vehicle models. This kind of approach gives good foundations to develop a robust system which is independent of the recorded signal. Furthermore, in some articles, the sample is too small, and the algorithms have been tested on unequal groups with just limited types of machine. What is more, in the current literature, there is a lack of research conducted on the impact of changing the sound compression. Further research is needed to develop accurate and efficient methods for automatic engine type identification using acoustic signals.

## 3. Architecture of Proposed System

A typical identification system structure includes three stages. The first is signal recording, followed by parameter extraction and classification. A diagram of the method proposed by the authors of this paper is shown in Figure 1.

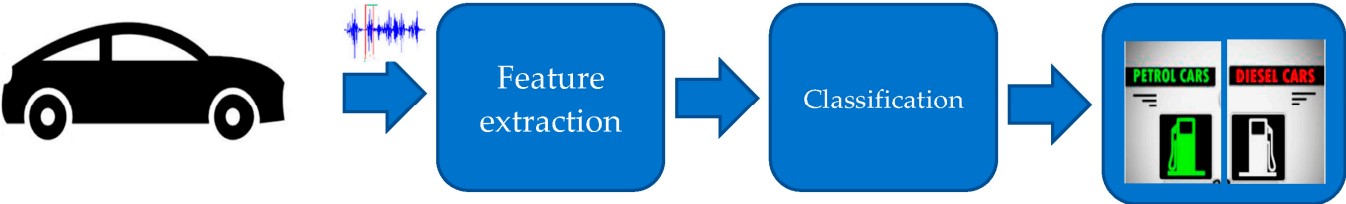

**Figure 1.** Block diagram of conventional of identification system.

### 3.1. Signal Recording

The experiment required creating a database with sounds of different vehicle engines. For this purpose, an Android-based smartphone was used; specifically, a Samsung S21 Ultra model manufactured by Samsung Electronics in Seoul, South Korea. The phone was equipped with a high-quality microphone designed for recording audio. The microphone was built into the phone and designed to capture high-quality sound in a variety of environments. To record the engine sounds, a freeware android Dyktafon app was used, which allowed for easy and convenient recording and saving of audio files. Sound signals were recorded with a sampling frequency of 44.1 kHz and in the WAV format, owing to the good quality of such a signal determined by the lossless format. Audio signals were recorded at different places, where the acoustics, surroundings and external factors varied. This significantly impacted the diversity of recorded sounds, which made constructing the entire system difficult. The dissimilarity of each audio signal in the database allows the designed signal to reflect real recording conditions with a high degree of probability. This enables answering the question of how the system would operate under real conditions. Figure 2 shows a test bench for recording sound samples.

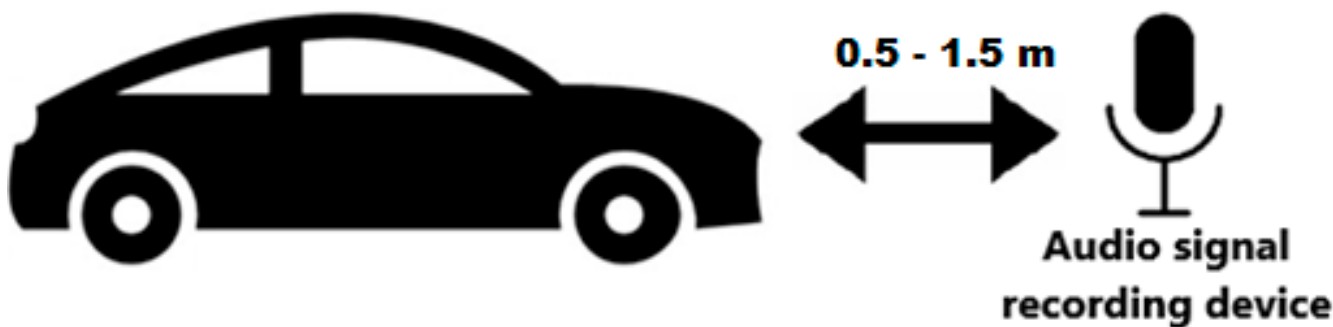

**Figure 2.** Test bench for recording sound samples.

Audio signals were recorded with the engine in idle run, that is, without any loads besides internal resistance. The recording device was positioned at 0.5–1.5 m from the vehicle bonnet. Most recordings were conducted at car dealerships, because obtaining such a large number of recordings with a satisfactory sampling frequency was a great challenge to the authors of this paper. Sound samples were recorded mainly in the morning due to the specificity of car dealership operation.

The collected database contains a total of 80 sound signals. The first 40 recordings originate from diesel engine vehicles, and the other 40 from petrol engine vehicles. Different vehicle models were recorded (approximately 60 in total). Table 1 presents synthetic dataset information, including the different types of machines used in recording.

**Table 1.** Dataset information.

| Vehicle Manufactures | Vehicle Model (Number) |
| --- | --- |
| AUDI | A3 (2) |
| | A4 (2) |
| | A5 (1) |
| | A6 (2) |
| | A8 (1) |
| BMW | 320d |
| | E46 (2) |
| | Seria 1 |
| | X5 |
| CITROEN | C1 |
| | C4 |
| | C5 |
| | Picasso |
| FIAT | Panda (3) |
| | Punto |
| FORD | Focus (3) |
| | Galaxy |
| | Mondeo |
| HYUNDAI | I20 (2) |
| | i40 |
| MAZDA | 6 |
| MERCEDES | 212 |
| | C |
| | CLC 200 |
| | s320 |
| MINI COOPER | Mini Cooper |
| MITSUBISHI | Pajero |

**Table 1.** *Cont.*

| Vehicle Manufactures | Vehicle Model (Number) |
|---|---|
| OPEL | Astra |
|  | Corsa |
|  | Insignia (2) |
|  | Meriva |
| PEUGEOT | 207 (2) |
|  | 2008 |
|  | 5008 |
|  | Boxer |
| RENAULT | Clio III |
|  | Megane |
|  | Clio |
|  | Scenic |
| ROVER | R75 |
| SAAB | 9-3 |
| SEAT | Altea |
|  | Exeo |
|  | Ibiza |
|  | Leon |
| SKODA | Octavia (2) |
|  | Rapid |
| SUZUKI | Swift |
|  | SX4 |
|  | Vitara |
| TOYOTA | Yaris |
| VOLKSWAGEN | Golf (3) |
|  | Passat (3) |
|  | Polo |
|  | Tiguan (2) |
| VOLVO | c30 |
|  | c60 |

A full specification of the recorded signals can be found in Tables A1 and A2 in Appendix A. In this paper, only sounds generated by the vehicle engine were used. In further work on the system, the author will focus on other sounds generated by vehicles, such as exhaust system.

*3.2. Feature Extraction*

The extraction and selection of features obtained from recorded signals is the work stage most important in terms of designing each identification system. The process is aimed at choosing such parameters of a recorded signal, so as to achieve features characteristic for each class of acquired sounds. The obtained descriptors will be used to define the target feature vector describing a given signal. The objective of processing sound signals with the use of the appropriate algorithm is to bring to light the distinguishing sound features of a given model. The feature extraction process involved employing the Matlab 2017b (The Mathworks Inc., Natick, MA, USA.) computation environment and defining a 12-element feature vector containing parameters defined within the frequency domain [39,40]: spectral Centroid, spectral Crest, Spectral Decrease, spectral Entropy, spectral Flatness, spectral Flux, spectral Kurtosis, spectral RolloffPoint, Spectral Skewness, spectral Slope, Spectral Spread, Pitch. The descriptors are more thoroughly described in [40–46]. A 15 ms Hamming window with a 5 ms overlap was used. The conducted experiments primarily utilized the audioFeatureExtractor function in Matlab [47]. The diagram of this function is graphically depicted in Figure 3. The list of parameters is shown in Table 2.

**Table 2.** List of extracted audio signal features.

| Parameter | Name |
|---|---|
| $C_1$ | Spectral Centroid |
| $C_2$ | Spectral Crest |
| $C_3$ | Spectral Decrease |
| $C_4$ | Spectral Entropy |
| $C_5$ | Spectral Flatness |
| $C_6$ | Spectral Flux |
| $C_7$ | Spectral Kurtosis |
| $C_8$ | Spectral RolloffPoint |
| $C_9$ | Spectral Skewness |
| $C_{10}$ | Spectral Slope |
| $C_{11}$ | Spectral Spread |
| $C_{12}$ | Pitch |

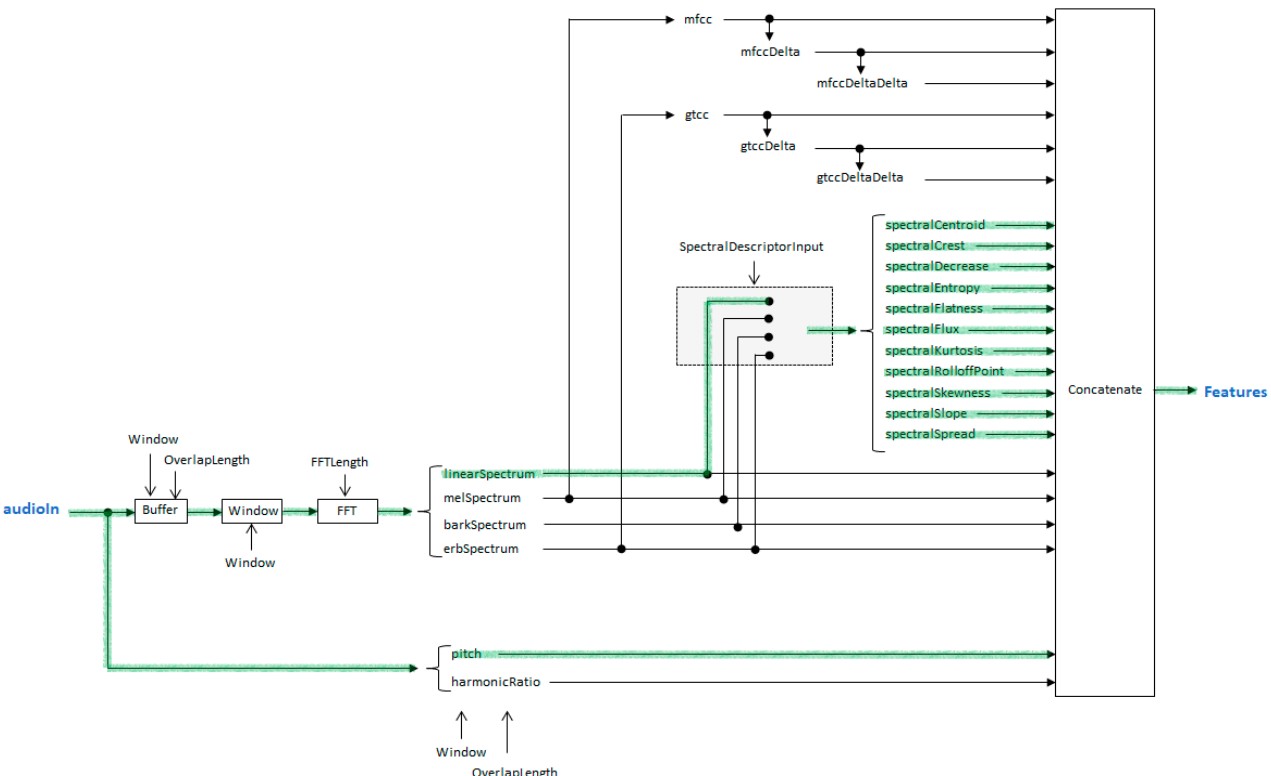

**Figure 3.** The diagram of the audioFeatureExtractor function.

*3.3. Selection*

The goal of applying feature selection techniques in machine learning is to find the best set of features that allows the building of optimized models of the studied phenomena [48]. Fisher score is one of the most widely used supervised feature selection methods. The key idea of the Fisher score is to find a subset of features such that in the data space spanned by the selected features, the distances between data points in different classes are as large as possible, while the distances between data points in the same class are as small as possible. The assessment involved employing the Fisher significance coefficient defined by the formula [49]:

$$S_{mn}(f) = \frac{|c_m - c_n|}{\sigma_m + \sigma_n} \tag{1}$$

where:

- $c_m$—$m$-class arithmetic mean

- $c_n$—$n$-class arithmetic mean
- $\sigma_m$—$m$-class standard deviation
- $\sigma_n$—$n$-class standard deviation

### 3.4. Classification

The classification stage involves generating predictions regarding objects from outside the training set, as based on input data. The classifier development procedure is divided into two phases. The first, called the "learning process", is responsible for creating a so-called "model" based on parameter values and classes. In addition, this process should also ensure the lowest possible classification error level. The next stage is determining classifier effectiveness through its testing by involving samples not participating in the learning process. In the current study, the preliminary structure of the vehicle class identification system was firstly examined using different machine learning techniques. The target structure of system was then determined by the conducted experiments which had the intent of achieving the best accuracy. The classification results were presented using the confusion matrix [50]. This is a simple cross-tabulation of the actual and recognized classes and allows easy calculation of the classifier parameters. The main indicator used to evaluate the proposed solution was accuracy [50].

$$Acc = \frac{TP + TN}{TP + TN + FP + FN} \tag{2}$$

## 4. Experiment

Constructing a system for identifying engine types based on the audio signals that they generate required, above all, conducting a preliminary analysis of the recorded signals to compare them. This was followed by selecting audio signal descriptors defined at the extraction stage. These experiments became the cornerstone for presenting assumptions related to the target system structure.

### 4.1. Observation of Studied Signals

The preliminary analysis of studied signals is based on presenting individual time waveforms and their spectra for the recorded sound samples. Figure 4 shows the waveform of an audio signal in a time domain, for diesel and petrol engines, respectively.

On comparing the audio signal time domain waveforms shown in Figure 4, a significant difference in the amplitude of the presented signal is noticeable. Both engines were recorded from the same distance of 0.5 m, thus minimizing the probability of distorting the obtained values. Red marks the signal envelope understood as an instantaneous amplitude value. The diesel engine amplitude ranges from –1 to 1, while the petrol engine ranges from −0.6 to 0.6 on average. Such a significant difference results from the design characteristics of the analyzed engine types.

Figure 5 shows examples of audio signal spectra for the recorded signals of the diesel and petrol engines, respectively. The tested engines have frequency components with the highest amplitude, ranging from 400 to 1400 Hz. The difference in low- and medium-frequency component amplitudes between the spectra is clearly noticeable. Spectrum amplitudes of the audio signal recorded for the petrol engine have considerably higher values relative to the diesel engine in the 0.5–8 and 12–15.5 kHz ranges.

### 4.2. Evaluation of Specific Audio Signal Features

Developing the target structure of the feature vector for the vehicle class identification audio signal is based on defining differences in the recorded signals. This goal is achieved through the feature selection process that can be treated as the problem of searching for a set of traits describing an object classified according to a certain evaluation criterion. Feature selection methods are usually composed of three elements (steps), namely feature subset generation, subset evaluation, and stop criterion. Basic statistical parameters, i.e., mean

value, standard deviation, and variance of a given parameter [51], were adopted as the evaluation criterion within the planned experiments. Tables 3 and 4 show the mean values of the extracted spectral parameters, calculated based on forty audio signals, assuming that two classes were defined for two engine types, respectively.

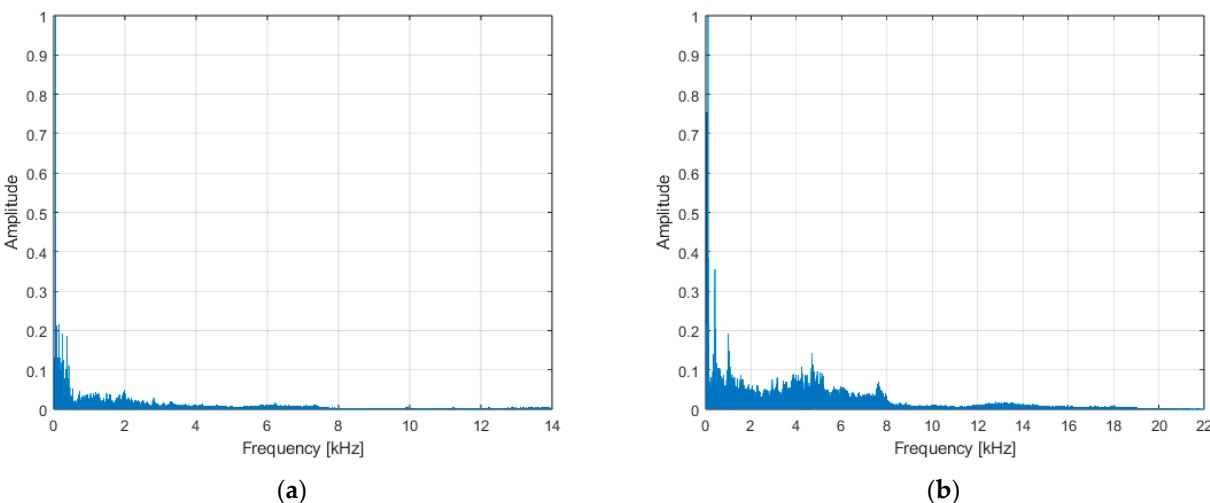

**Figure 4.** Audio signal waveform in a time domain recorded for (**a**) diesel engine and (**b**) petrol engine.

**Figure 5.** Spectrum of the audio signal recorded for (**a**) diesel engine and (**b**) petrol engine.

**Table 3.** List of extracted audio signal features with obtained numerical results for the diesel engine.

| Parameter | Mean | Standard Deviation | Variance |
|---|---|---|---|
| $C_1$ | 1346.303 | 354.309 | 125,535.063 |
| $C_2$ | 766.1 | 94.678 | 8963.879 |
| $C_3$ | 0.054 | 0.005 | $2.777 \times 10^{-5}$ |
| $C_4$ | 0.580 | 0.030 | $9.279 \times 10^{-4}$ |
| $C_5$ | 0.041 | 0.014 | $2.033 \times 10^{-4}$ |
| $C_6$ | 0.004 | $9.489 \times 10^{-4}$ | $9.004 \times 10^{-7}$ |
| $C_7$ | 18.975 | 6.581 | 43.304 |
| $C_8$ | 7536.43 | 3395.796 | 11,531,431.954 |
| $C_9$ | 3.745 | 0.667 | 0.445 |
| $C_{10}$ | $-3.376 \times 10^{-9}$ | $5.926 \times 10^{-10}$ | $3.512 \times 10^{-19}$ |
| $C_{11}$ | 2847.529 | 607.579 | 369,152.539 |
| $C_{12}$ | 62.784 | 50.028 | 2502.791 |

**Table 4.** List of extracted audio signal features with obtained numerical results for the petrol engine.

| Parameter | Mean | Standard Deviation | Variance |
|---|---|---|---|
| $C_1$ | 2049.129 | 222.143 | 49,347.569 |
| $C_2$ | 146.823 | 42.035 | 1766.918 |
| $C_3$ | 0.026 | 0.004 | $1.668 \times 10^{-5}$ |
| $C_4$ | 0.732 | 0.02 | $4.192 \times 10^{-4}$ |
| $C_5$ | 0.056 | 0.008 | $6.431 \times 10^{-5}$ |
| $C_6$ | $3.622 \times 10^{-4}$ | $1.669 \times 10^{-4}$ | $2.788 \times 10^{-8}$ |
| $C_7$ | 11.785 | 2.256 | 5.088 |
| $C_8$ | 7081.120 | 336.166 | 113,007.696 |
| $C_9$ | 2.675 | 0.298 | 0.089 |
| $C_{10}$ | $-3.034 \times 10^{-10}$ | $6.994 \times 10^{-11}$ | $4.892 \times 10^{-21}$ |
| $C_{11}$ | 2871.476 | 178.671 | 31,923.415 |
| $C_{12}$ | 280.566 | 126.810 | 16,080.764 |

The diversity of mean values for specific engine types is the main aspect that reveals the usefulness of a given parameter within the process of designing a vehicle class identification system. Standard deviation is an additional, equally important factor. It is the basic measure of the variability in the values of defined parameters. In the case of large values of this parameter for a given feature, the parameter is rejected because its numerical values are too scattered within one class, which leads to the analyzed feature being hardly repeatable for the class in question.

The extracted features were empirically divided into two subgroups, based on determined parameters. Group 1 ($C_2, C_3, C_4, C_5, C_6, C_7, C_9, C_{10}$) contains descriptors that satisfy the assumed conditions, and the authors believe they can be employed to distinguish different engine types. Mean values reach values clearly different relative to the two classes and standard deviations are characterized by low values, which indicates their strong concentration around the mean value. The remaining features comprise Group 2 ($C_1$, $C_8, C_{11}, C_{12}$). Accordingly, they do not exhibit good discriminatory properties in relation to the issue under consideration. The authors conclude that the probability of obtaining the same values for two different engine types is too high due to similar mean values of the individual features that distinguish them as parameters comprising the vector of the features describing an audio signal [52].

In the further part of the article, the Fisher measure was also used. Known as "Fisher information", these are statistical measures used to quantify the amount of information that an observable random variable carries about an unknown parameter of interest. They are named after the statistician, Ronald Fisher, who introduced them in the early 20th century.

### 4.3. Assessment of Qualified Parameters

The process of preliminary parameter value change assessment enabled selecting a feature vector consisting of 8 descriptors. Figure 6 shows a waveform of value changes in selected parameters in order to find differences between them.

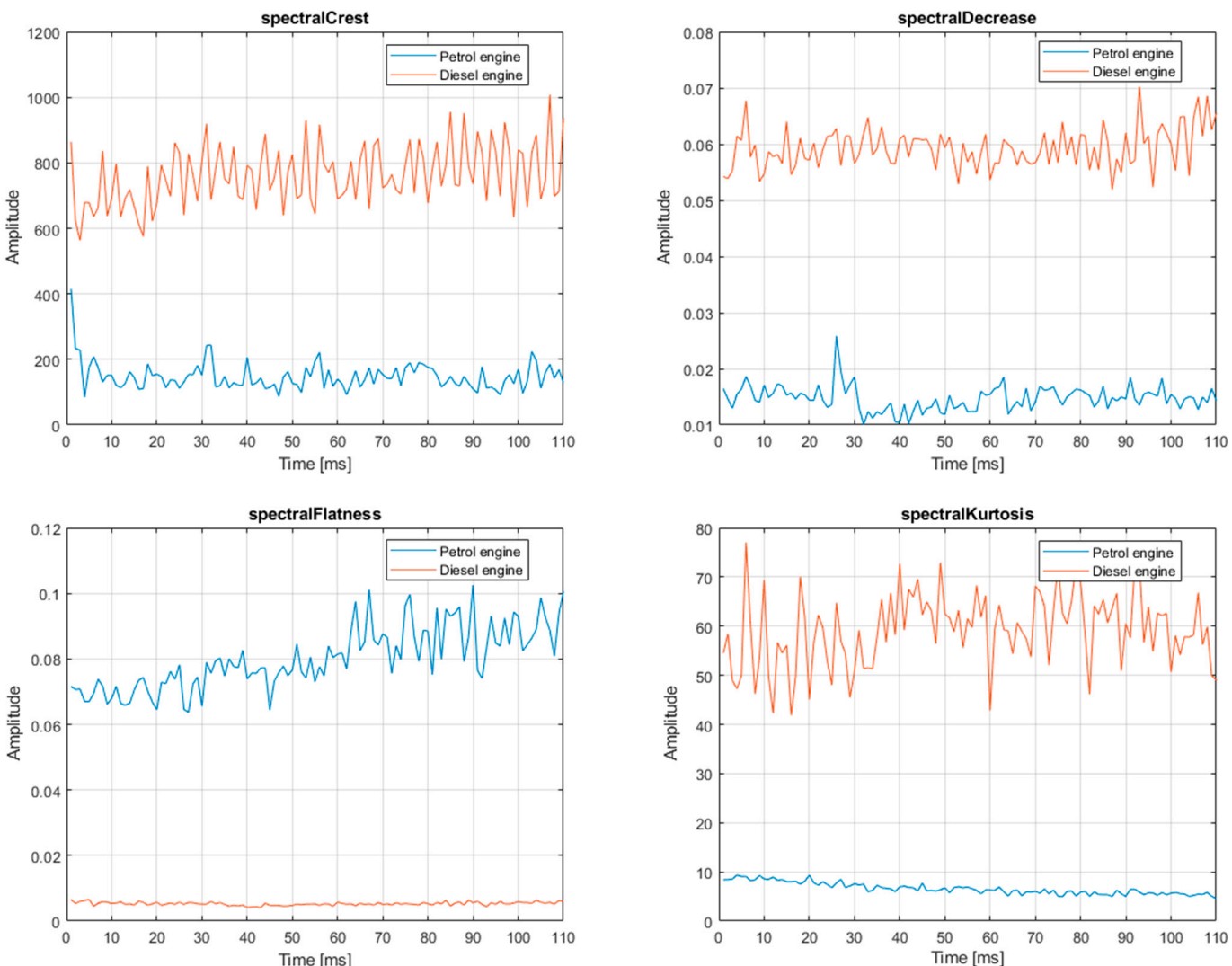

**Figure 6.** Waveform of value change in selected descriptors for the petrol and diesel engines.

A significant difference in the mean values between two vehicle classes was noticed based on the data from Tables 3 and 4. Moreover, the standard deviations taken by this descriptor within a given class are at an acceptable level, despite high set variability. According to the authors, the Spectral Crest descriptor can become an important information carrier when constructing a vehicle identification system. Supported by other descriptors, it will constitute the ultimate vector of the features describing an audio signal. Spectral Decrease is characterized by very low numerical values. When observing the time waveform shown in Figure 3 and the data from Tables 1 and 2, it is noticeable that the mean value for the diesel engine is approximately 0.053 and is halved for the petrol engine. Its standard deviation is low, which increases its credibility as a feature distinguishing two engine types. In addition, when looking at this variance, one can conclude that given values are stable over time. The Spectral Flatness feature is similar in terms of value to the c3 feature, namely, spectral decrease. However, the difference between the mean values is no longer twofold. Analyzing the remaining numerical values of this parameter enables a conclusion that this descriptor can

be used to identify a vehicle class with high probability. In addition, the waveform shown in Figure 6 demonstrates the stability in the value of this parameter for the diesel engine.

### 4.4. Results of Recognition

The designed system, taking only 8 extracted descriptors into account, was initially subjected to classification. The classification model is based on a training set, and the assessment of its effectiveness has been verified based on a test set. Audio signals were divided in the 75%-to-25% proportion into the training and validation sets [53], respectively. In consequence, 30 training sounds and 10 test sounds were obtained for each engine type. The abilities of the 12-dimensional vectors based on spectrum of sound signal, were examined with standard machine learning methods. Table 5 lists the effectiveness of individual classifiers as measured by accuracy. The conducted experiments demonstrate that the most accurate results of recognition were achieved with linear support vector machine SVM [54].

**Table 5.** List of the effectiveness of individual classifiers.

| Algorithm | Accuracy [%] |
|---|---|
| Fine Tree | 68.8 |
| Medium Tree | 68.8 |
| Coarse Tree | 73.8 |
| Linear Discriminant | 86.2 |
| Quadratic Discriminant | 82.5 |
| Logistic Regression | 81.2 |
| Gaussian Naive Bayes | 71.2 |
| Kernel Naive Bayes | 67.5 |
| Linear SVM | 91.7 |
| Quadratic SVM | 80 |
| Cubic SVM | 77.5 |
| Fine Gaussian SVM | 61.3 |
| Medium Gaussian SVM | 81.2 |
| Coarse Gaussian SVM | 78.8 |
| Fine KNN | 76.2 |
| Medium KNN | 73.8 |
| Coarse KNN | 50 |
| Cosine KNN | 76.2 |
| Cubic KNN | 73.8 |
| Weighted KNN | 77.5 |
| Boosted Trees | 50 |
| Bagged Trees | 77.5 |
| Subspace Discriminant | 83.8 |
| Subspace KNN | 62.5 |
| RUSBoosted Trees | 50 |
| Narrow Neural Network | 73.8 |
| Medium Neural Network | 83.8 |
| Wide Neural Network | 78.8 |
| Bilayered Neural Network | 82.5 |
| Trilayered Neural Network | 76.2 |
| SVM Kernel | 57.5 |
| Logistic Regression Kernel | 58.8 |

The trained classifier misclassified two audio signals out of twenty, one for the diesel engine, and one for the petrol engine. The validation matrix for the linear SVM is presented in Figure 7. It demonstrates that the system recognizes vehicles with diesel and petrol engines with 90% accuracy.

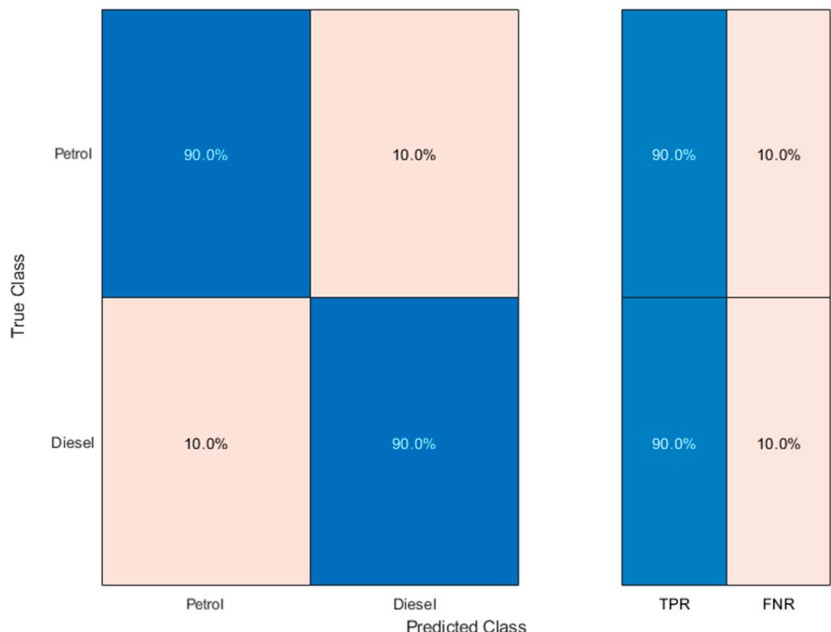

**Figure 7.** Classifier effectiveness for the petrol and diesel engines for linear SVM.

Because of the limited data sample, the authors applied the cross-validation method in order to reliably assess the proposed method for recognition. In this variant, the feature vector based on the descriptors will be divided into k number of equal subsets. Each of these will be successively classified as a test set, and the combination of the other ones as a training set. This will be followed by a k-fold analysis of each of them and the obtained results will be averaged in order to obtain effectiveness values of the final system [50]. An advantage of the cross-validation system is its accuracy and that it does not employ data for testing. Classifiers for nine different cases, starting with $k = 2$ and ending with $k = 10$, were trained under this variant. The effectiveness of all tested methods is compared in Figure 8. Classifier effectiveness relative to the adopted 10-fold cross-validation is presented in Figure 9.

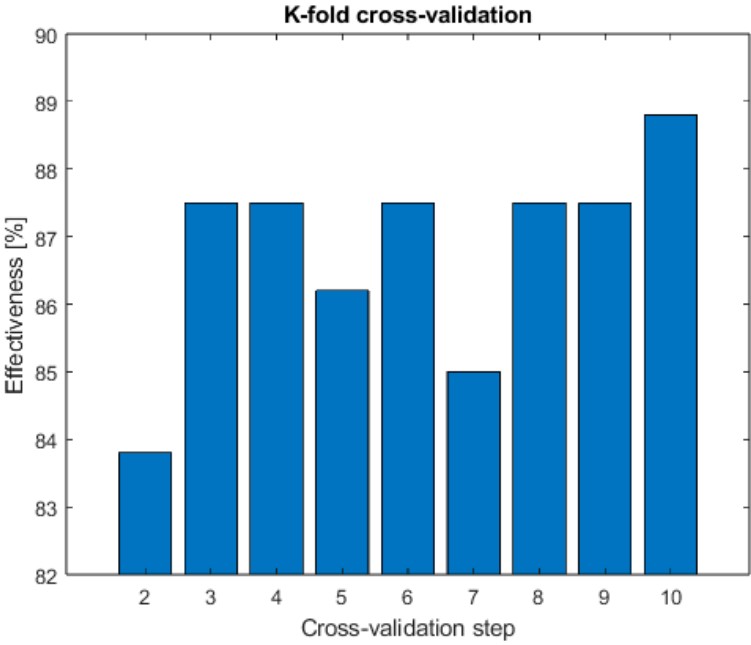

**Figure 8.** Classifier effectiveness relative to the adopted k-fold cross-validation.

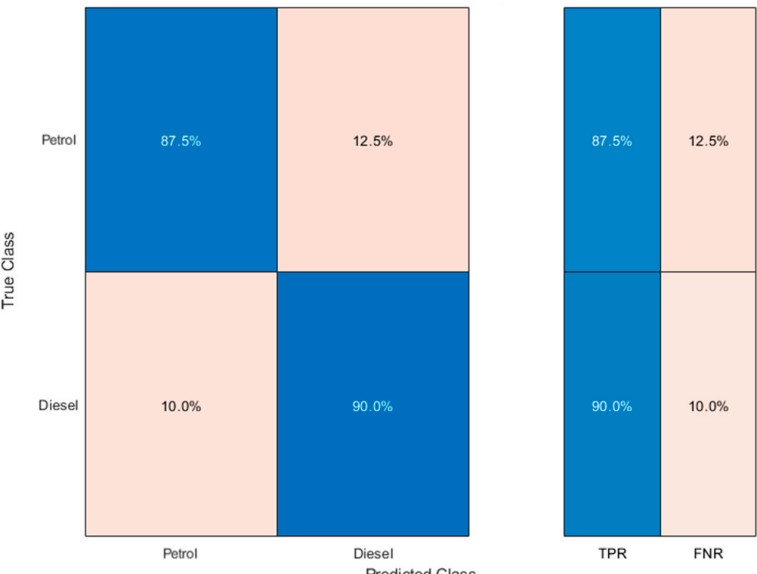

**Figure 9.** Classifier effectiveness relative to the adopted 10-fold cross-validation.

*4.5. Feature Selection*

The feature importance selection algorithm was applied in order to reduce the number of parameters used for classification, while simultaneously enhancing the classification result. It can possibly indicate synergy among features, as well the synergy of features with the SVM classifier, and the applied algorithm of their selection [51]. Three groups can be distinguished among object-describing features. These are relevant, irrelevant, and redundant. The first group contains features good at "distinguishing" between classes and which improve classification algorithm effectiveness. Irrelevant features are those wherein the value of which are random in each class. They usually do not lead to improved classification effectiveness or even worsen it. The third feature group contains those in which their roles can be taken over by other features. As part of the research, the authors applied one of the so-called "ranking methods", the essence of which is an attempt at finding relevant features, taking their assessment measure into account.

Figure 10 shows a so-called "feature ranking" or, in other words, values of the Fisher measure for individual descriptors. Descriptors describing spectral slope ($C_{10}$) and crest ($C_2$) stand out the most, since their Fisher measure values exceed 4.5.

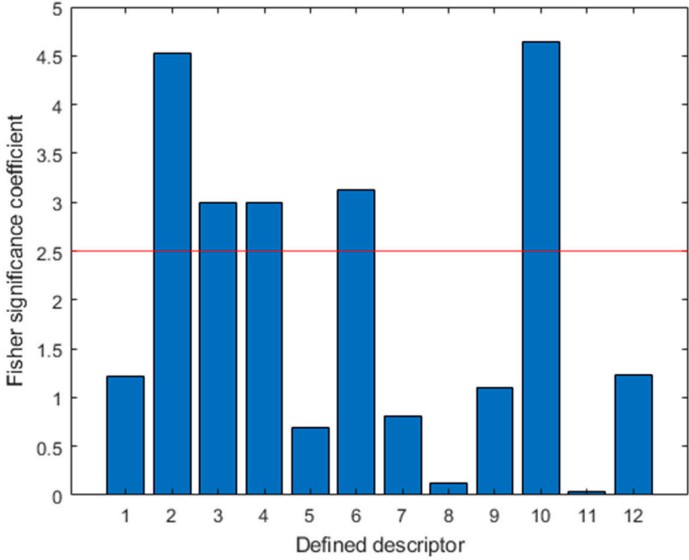

**Figure 10.** Fisher measure values for the defined descriptors (the red line shows mean value).

Another classifier that exhibits effectiveness of 80% was developed based on five se-lected descriptors and a linear SVM algorithm. Herein, narrowing down the number of descriptors to two best at distinguishing between both engine types enabled obtaining identification effectiveness of 85%, which is a small difference. A confusion matrix in the case of using two descriptors is presented in Figure 11 below.

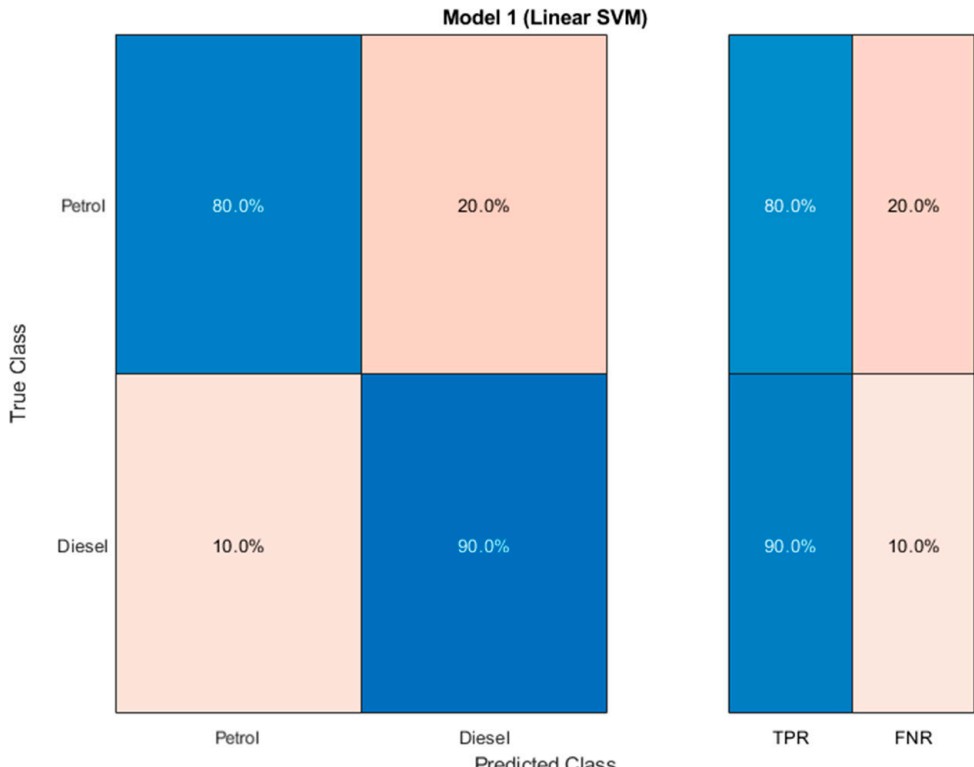

**Figure 11.** Validation validity matrix for two descriptors.

*4.6. Evaluation the Database*

Wide diversity within the dataset and the unsupervised process of signal recording brought about a situation whereby the size of the learning database matters, as it has an impact on performance and its management methodology. In this work, we recognized that different environmental conditions may affect such sensitive data recordings and the final results of identification.

The research into database evaluation divided it into different learning and testing stages so as to assess the size of database for the recognition quality. The sound recordings were divided into five subgroups:

- 1/2 is a training set, and 1/2 is a validation set
- 3/5 is a training set, a 2/5 is a validation set
- 7/10 is a training set, a 3/10 is a validation set
- 4/5 is a training set, a 1/5 is a validation set
- 9/10 is a training set, a 1/10 is a validation set

*4.7. Evaluation the Impact of Lossy Compression Sound Signal*

The main objective of these experiments was to assess the impact of changing the sound compression on the effectiveness of the solution proposed by the authors. Lossy compression removes details irreversibly. In the MP3 files, the compression algorithm is based on the range of human hearing, and sound that is inaudible or insignificant to the human ear is removed from the file.

Figure 12 below shows a summary of system operation efficiency for the different divisions into training and test data.

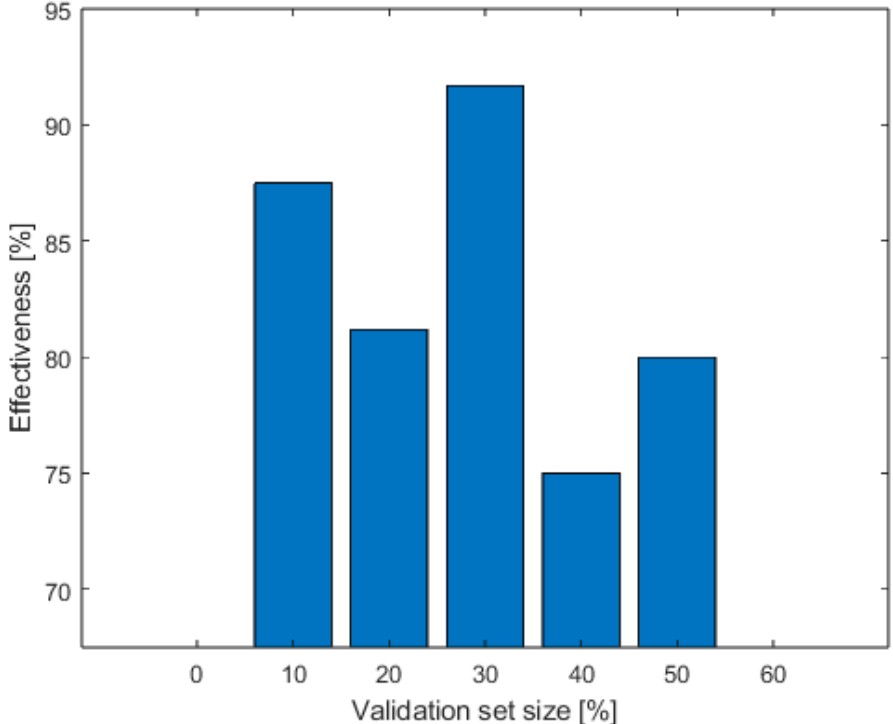

**Figure 12.** System operation efficiency depending on training and test database sizes.

People can hear frequencies within the 20 Hz–20,000 Hz range, but the human ear is most sensitive to a smaller range, generally given as 100 Hz to about 6 kHz. Therefore, in theory, any quiet content in the low-end and high-end can also be removed without a noticeable impact on the overall sound quality [55]. Unfortunately, this compression changes the spectrum of the analyzing sound that is the base for the calculated features. Table 6 shows mean values of extracted spectral descriptors determined by the authors with the use of the feature extraction algorithm for the diesel engine sound recorded in .wav and .mp3 formats.

**Table 6.** List of extracted audio signal features with obtained numerical results for the diesel engine.

| Parameter | .wav File | .mp3 File |
|-----------|-----------|-----------|
| $C_1$ | 1349.678 | 1346.303 |
| $C_2$ | 763.883 | 766.1 |
| $C_3$ | 0.055 | 0.054 |
| $C_4$ | 0.579 | 0.580 |
| $C_5$ | 0.025 | 0.041 |
| $C_6$ | 0.004 | 0.004 |
| $C_7$ | 18.460 | 18.975 |
| $C_8$ | 7636.798 | 7536.43 |
| $C_9$ | 3.697 | 3.745 |
| $C_{10}$ | $-3.076 \times 10^{-9}$ | $-3.376 \times 10^{-9}$ |
| $C_{11}$ | 2853.089 | 2847.529 |
| $C_{12}$ | 68.314 | 62.784 |

Validation validity matrix for this variant is shown in Figure 13.

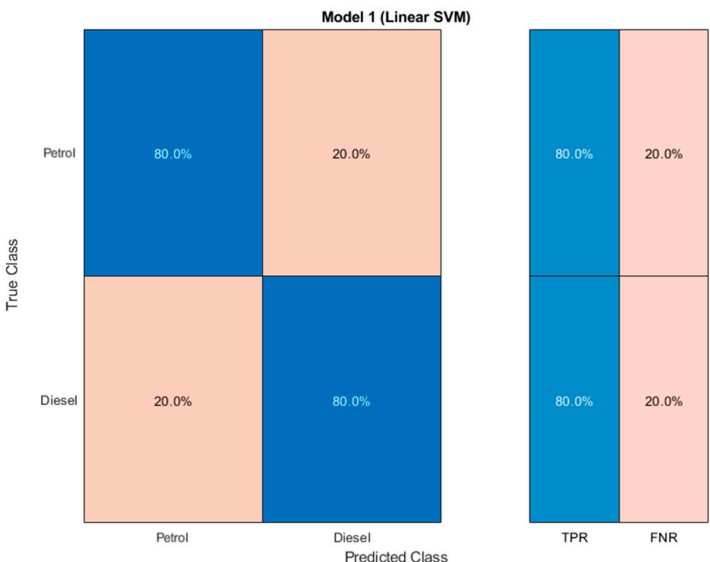

**Figure 13.** Validation validity matrix for mp3 sounds.

*4.8. Assessing System Operation Efficiency Based on a Selected Audio Signal Type*

The last experiment assumes classifier training based on signals involving only vehicle ignition type input. To this end, Audacity software was used to manually separate the sound of an engine running in idle speed from the signal containing its start-up. Next, the features were extracted. The $C_2$ $C_4$ $C_6$ $C_7$ $C_9$ $C_{12}$ descriptors were chosen for classifier training. The validation matrix is shown in Figure 14.

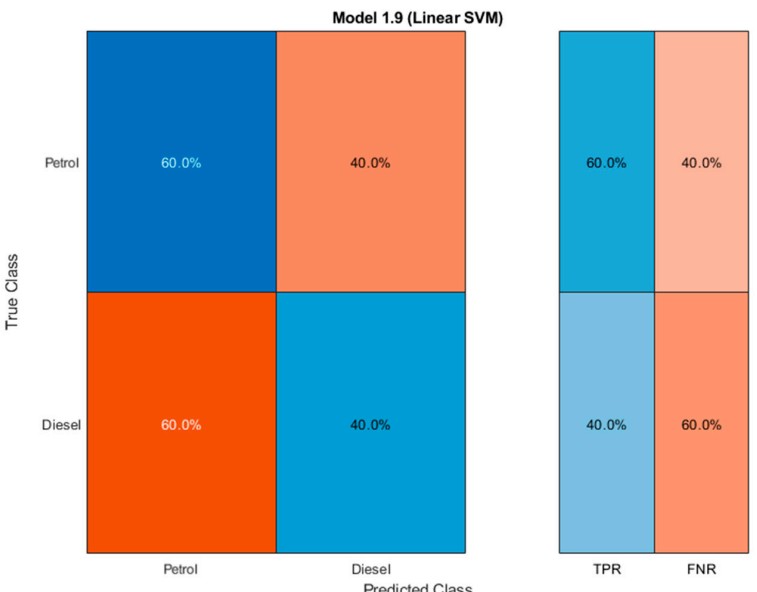

**Figure 14.** Validation validity matrix for audio signals containing start-up.

## 5. Discussion

The research proposed by the authors focused on recognize a type of engine (petrol or diesel), regardless of type of machine. The most important stage of this research is to develop the architecture of the recognizing system.

The execution of this task required, above all, characterizing and recording a sound database. The research utilized real recordings acquired in cooperating car dealerships within Poland. The sound database recorded by the authors contains 80 various audio signals, equally divided into that generated by diesel and petrol engines. The second

stage of research focused on feature extraction stage, taking into account the evaluation of specific audio signal features. Matlab software using the audioFeatureExtractor function was utilized for this purpose. Next, the authors empirically selected the parameters that they believed to hold the ability to distinguish between classes. To this end, the authors employed basic statistical parameters for analysis, and executed comparative analyses of changes in the value of a given feature in the time domain. Accordingly, eight of 12 parameters defined at the descriptor extraction stage were pre-qualified. Selected features and different machine learning methods were then used to choose a linear SVM classifier characterized by an effectiveness level of 90%, where the sound database was divided in a 3:1 ratio to training and test data. We used the accuracy indicator and the confusion matrix to evaluate the results.

Because the dataset of the experiment is limited, the cross-validation method was then applied. An advantage of the cross-validation system is its accuracy and that it does not employ data for testing. The conducted experiments also involved defining the number of tests aimed necessary for assessing the solution proposed by the authors. The results obtained using the test method showed that the classifier was characterized by the lowest effectiveness of 83.8% for k = 2 and the highest for k = 10. Effectiveness reductions relative to their predecessors were noted for validation steps of k = 5 and k = 7. A more detailed analysis was subsequently conducted for two randomly chosen cases. In the first, the set was divided into k = 5 subsets, and into k = 10 subsets in the second. Accordingly, an identification system designed based on five-fold analysis and a linear SVM algorithm was characterized by an effectiveness of 86.2%. In the case of a five-fold validation, we noted that petrol engines are characterized by lower identification effectiveness (82.5%) relative to diesel engines (90%). In the second case, which assumed a 10-fold validation, the system exhibited higher effectiveness at a level of 88.8%. Figure 8 shows the error percentage for this system testing variant.

Another problem assumed by the authors is evaluation of the database. The diversity in recording signals is very wide. Although there are different approaches to collecting data for machine learning models, and it ultimately depends on the specific goals of the project, in this particular study, the author chose to collect the data in an uncontrolled environment to simulate real-world conditions, where the engine sound would be mixed with other sounds and background noise. The aim was to make the model more robust and adaptable to application in different environments and situations. An analysis of data division into training and test indicated that the first variant assumed an equal set division (50% of training and test data each). Eight signals out of forty were incorrectly classified, five as petrol engine and three as diesel engine. Another classifier obtained the lowest effectiveness of all at 75%. In this case, five petrol engines and three diesel engines were incorrectly classified out of all 36 in the validation set. The third subgroup had a 70–30% set ratio. The classifier developed in this proportion demonstrated the highest effectiveness of all proposed within this diploma thesis, at a level of 91.7%. Only two signals in 24 were misclassified and belonged to the petrol engine group. The fourth division exhibited a similar effectiveness to the first one, amounting to 81.2%. In this case, one of the diesel engine sounds and two of the petrol engine sounds were incorrectly classified in the test database containing 16 sounds in total. The last classifier had the smallest validation set of all, because it only consisted of eight of these sounds. One incorrectly classified engine in such a small set translated to a system effectiveness of 87.5%.

The subsequent research we undertook is related to evaluating lossy sound signal. The main objective of the experiments was to assess the impact of changing the sound compression on the effectiveness of the solution we proposed. In undertaking this work, we discovered that when the format of the analyzed file is changed to a lossy one such as mp3, the obtained numerical values of extracted descriptors do not significantly differ from each other. However, after network training under the same assumptions, we saw that system effectiveness decreased from 90% to 80%. With regard to the test results, it can be concluded that the format of the audio signal does not significantly affect the effectiveness of the system. To conclude, the proposed system is independent of the format recorded audio signal.

Another area of research that we undertook is related to the limited set of recorded signals. Herein, we observed that deterioration in identification results was observed when assessing system operation effectiveness based on a selected audio signal type. A system trained on sounds of engine ignition type alone guarantees an effectiveness of only 50%. Such a classification level is unsatisfactory since a random hit has the same probability statistically.

## 6. Conclusions

The objective of this research was to construct a motor vehicle engine type identification system based on the parametric analysis of an audio signal.

The study was conducted using feature engineering techniques based on frequency analysis for the generation of sound signal features. The discriminatory ability of feature vectors was evaluated using different machine learning techniques. In order to test the robustness of the proposed solution, the authors executed a number of system experimental tests, including different work conditions for the proposed system. The study involved five basic experiments. The first assumed training a classifier based on the entire database of audio signals with the use of a cross-validation method. Another variant was based on training a classifier using a database of sounds converted to the mp3 format. A further experimental test variant assumed the application of a feature selection algorithm (Fisher significance coefficient). The fourth experimental test variant assumed classifier training based on signals containing only car ignition type. The last of the variants assumed assessing identification effectiveness, taking into account data division into training and test. The results show that our system achieved performance of 91.7% in terms of accuracy.

Most of the studies presented in the literature describe the identification of a type of machine defined as car, bus, truck, or motorcycle [27]. The study reported by [28], in contrast, identified the car model. The research proposed by us focused on type of engine (petrol or diesel), regardless of machine type. This study is comparable to that of [28], but the aim of the study is different. To our knowledge, our study is a first [34,35], so the possibility to compare with other results is limited.

The presented material extends the findings discussed in the most comparable paper [34,35]. Although we reached lower overall accuracy: 91.7%, we used in our experiments a more numerous and diverse database. In [34], the system works on smaller dataset. It identifies six different automotive engine sounds from six different vehicle makes with two types of engine, petrol and diesel, respectively. In our dataset, we used recordings of vehicles built by 21 different vehicle manufacturers, as well as 57 different car models. The greater diversity in our work as compared to previous studies in the literature lies in the greater diversity in the database as it includes the offerings of different car manufacturers as well as different car models. This kind of approach gives good foundations for developing a robust system that is independent of recorded signal, vehicle make, and model. In [35], the authors investigated four different automotive engine sounds from just four different makes of vehicle with two types of engine (V6 and V8). The results give better accuracy probably because the vehicle type is limited, so the sound is more repeatable.

Besides, it turned out that experiments conducted with the use of changing sound compression can give good results. With regard to the test results, it can be concluded that the format of the audio signal does not significantly affect the effectiveness of the system. To conclude, the proposed system is independent of the format of the recorded audio signal.

Research in this field can be continued by expanding the classes of analyzed vehicles, e.g., trucks or motorcycles, and recording using different "acoustic signatures" for different recording places. The results of this research can be implemented in practice in various ways. For example, the proposed system can be integrated into intelligent transportation systems to improve traffic management, enhance vehicle safety, and reduce environmental pollution. The system can also be used in automotive service centers to quickly and accurately identify engine types. What is more, our system could be used, e.g., in preventing misfuelling diesel to petrol engines or petrol to diesel engines. This kind of system can be implemented in

petrol stations to recognize the car based on the sound of the engine. A system that merely identifies the vehicle visually does not provide this opportunity.

**Author Contributions:** Conceptualization, E.M.-Z.; data curation. E.M.-Z. and M.M.; formal analysis, E.M.-Z. and M.M.; investigation, M.M.; methodology, E.M.-Z.; resources, M.M.; software, M.M.; validation, M.M.; visualization, M.M.; supervision, E.M.-Z.; writing—original draft preparation, E.M.-Z.; writing—review and editing, E.M.-Z. All authors have read and agreed to the published version of the manuscript.

**Funding:** This work was financed/co-financed by Military University of Technology under research project UGB 865.

**Data Availability Statement:** The data presented in this study are available on request from the corresponding author. The data are not publicly available due to privacy and to prevent mass dissemination of the collected data, including the sounds of vehicles recorded during the study.

**Conflicts of Interest:** The authors declare no conflict of interest.

## Appendix A

**Table A1.** Signal characteristics for diesel engine vehicles.

| Car Model | Engine Capacity [L] | Signal Duration [s] |
|---|---|---|
| Audi A4 | 1.90 | 20 |
| Alfa romeo 159 | 2.00 | 18 |
| Audi A3 | 1.90 | 16 |
| Audi A6 | 3.00 | 12 |
| Audi A8 | 4.00 | 17 |
| Bmw 320d | 2.00 | 21 |
| Bmw E46 | 2.00 | 16 |
| Bmw Seria 1 | 2.00 | 23 |
| Bmw X5 | 3.00 | 21 |
| Citroen C5 | 2.20 | 15 |
| Citroen xantia | 2.10 | 17 |
| Ford Focus | 1.60 | 18 |
| Ford Focus | 1.60 | 10 |
| Ford Galaxy | 2.00 | 10 |
| Ford Mondeo | 2.00 | 12 |
| Hyundai i40 | 1.70 | 14 |
| Mazda 6 | 2.00 | 11 |
| Mercedes 212 | 3.00 | 11 |
| Mercedes C | 2.70 | 13 |
| Mercedes CLC 200 | 2.00 | 13 |
| Mercedes s320 | 3.20 | 14 |
| Mitsubishi Pajero | 2.50 | 14 |
| Opel Insignia | 2.00 | 13 |
| Peugeot 2008 | 1.40 | 13 |
| Peugeot 5008 | 2.00 | 14 |
| Peugeot Boxer | 2.50 | 11 |
| Renault Clio III | 1.50 | 22 |
| Renault Megane | 2.00 | 24 |
| Saab 9-3 | 1.90 | 31 |
| Seat Leon | 1.90 | 12 |
| Skoda Octavia | 1.90 | 10 |
| Skoda Rapid | 1.40 | 11 |
| Toyota Avensis | 2.00 | 11 |
| Volkswagen Golf | 1.90 | 12 |
| Volkswagen Passat | 2.00 | 10 |
| Volkswagen Passat | 2.00 | 11 |
| Volkswagen Passat | 1.90 | 11 |
| Volkswagen Tiguan | 2.00 | 10 |
| Volvo c30 | 2.40 | 12 |
| Volvo V60 | 2.00 | 12 |

**Table A2.** Signal characteristics for petrol engine vehicles.

| Car Model | Engine Capacity [L] | Signal Duration [s] |
|---|---|---|
| Audi A3 | 2.0 | 25 |
| Audi A4 | 3.0 | 35 |
| Audi A5 | 1.8 | 21 |
| Audi A6 | 2.8 | 27 |
| Bmw E46 | 2.0 | 20 |
| Chevrolet Aveo | 1.2 | 22 |
| Citroen C1 | 1.0 | 23 |
| Citroen C4 | 1.6 | 18 |
| Citroen Picasso | 1.6 | 19 |
| Fiat 500 | 1.4 | 7 |
| Fiat Panda | 1.2 | 9 |
| Fiat Panda | 1.2 | 15 |
| Fiat Panda | 1.2 | 16 |
| Fiat Punto | 1.4 | 8 |
| Ford Focus | 1.6 | 15 |
| Hyundai i20 | 1.2 | 23 |
| Hyundai i20 | 1.4 | 19 |
| Mini Cooper | 1.6 | 9 |
| Opel Astra | 1.4 | 7 |
| Opel Corsa | 1.4 | 18 |
| Opel Insignia | 1.8 | 16 |
| Opel Meriva | 1.4 | 38 |
| Peugeot 207 | 1.4 | 32 |
| Peugeot 207 | 1.4 | 13 |
| Renault Clio | 1.2 | 13 |
| Renault Scenic | 1.6 | 19 |
| Rover R75 | 1.8 | 17 |
| Seat Altea | 1.6 | 19 |
| Seat Exeo | 1.8 | 8 |
| Seat Ibiza | 1.2 | 27 |
| Seat Leon | 1.6 | 15 |
| Skoda Fabia | 1.2 | 6 |
| Suzuki Swift | 1.2 | 19 |
| Suzuki SX4 | 1.6 | 11 |
| Suzuki Vitara | 1.4 | 13 |
| Toyota Yaris | 1.3 | 22 |
| Volkswagen Golf | 2.0 | 26 |
| Volkswagen Golf | 1.4 | 12 |
| Volkswagen Polo | 1.4 | 9 |
| Volkswagen Tiguan | 1.4 | 13 |

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
