# Peer review of "Classification of Engine Type of Vehicle Based on Audio Signal as a Source of Identification"

_electronics, doi:10.3390/electronics12092012_

Round 1

Reviewer 1 Report

Authors contributions:

The authors have presented a method for vehicle class identification based on a recorded sound signal. In this case, vehicle class is construed as the type of engine the vehicle is fitted with. Internal combustion engines can be divided by different criteria; however, this study is focused on classifying them based on ignition method that determines the type of consumed fuel. Such systems can support intelligent transportation systems through employing a sound signal as a medium carrying information on the type of car moving along a road.

I have some reviewer notes:

Abstract: The problem is not well defined. At the end of this part, you have to describe how your results can be implemented in practice.

Introduction. The aim of this study is not clearly defined.

Related works. At the end of this part, you have to summarize what are the problems with the known solutions in this study area. Also, what are the task that you have to solve.

Line 113. “Samsung S21 Ultra phone”. Better presentation is Model (Manufacturer, City, Country). Also, give more details about technical specification of the microphone used.

3.1.Signal recording. The automobile has many other parts except engine. You have to explain their influence.

3.2.Feature extraction. What method for feature extraction do you use? How many features are used for Training, validations and testing of the classifier?

Line 142. Matlab 2017b (The Mathworks Inc., Natick, MA, USA.). It will be better description.

Line 156. I agree that LSVM is good classification algorithm, but what about other possible algorithms? The assessment criteria have to be described here.

Figure 7. You have to describe why the points are connected. You can present the figure as bar chart.

Line 276. It will be good to describe all audio formats that you use in Material and methods part. Not here.

Line 307. Fisher measures have to be described in Material and methods part.

Discussion part. You have to compare your results with minimum 3 other papers.

Conclusion part. How this work improves the known solutions in this study area? How your esults can be implemented in practice?

I have some suggestions:

Make better description of hardware and software that you use. Improve the structure of your paper. Some descriptions have to be in Material and methods part.

Reviewer 2 Report

This paper was very well written and organized, making your individual steps easy to follow. I particularly like the way you have structured each section and transitioned between ideas. However, I have some major concerns regarding the design of your experiment. These issues are detailed below. Best wishes and good luck with your future research.

-There is some confusion as to the primary goal of your study. You mention classifying engine type, which would suggest you are using a network capable of multi-class labelling (V6, V8, etc.). However, it appears you are only performing a binary classification (petrol vs diesel), which does not become apparent until the end of the first section. I suggest emphasizing this objective earlier in the text.

-Is there a reason you chose to use an SVM? Recent studies involving binary classification have demonstrated that deep neural networks, with ReLU activation functions in hidden layers and sigmoid functions in output layers, outperform SVMs (significantly in some cases).

-You mention the collection of data as one of the primary contributions of your study. Are these samples publicly available?

-It appears Line 77 has not been translated into English.

-You reference the study by Sidiropoulus [10] and report they were able to identify car model using an MLP with an accuracy of 98%. It is unclear how your study improves on their work, as you report an accuracy of only 91%. Also, could you not determine the engine type given the brand, make, and model of the vehicle?

-How does your work improve on that of references [13] and [14]?

-You collected your data in an uncontrolled environment and argue this stochastic noise makes the samples more robust. I would argue this approach is not desirable for machine learning. In contrast, you should be collecting the highest-quality samples available for training and performing pre-processing steps to eliminate noise where possible. Otherwise, how do you know the network is not learning from the background noise rather than the signal?

-In the introduction, you describe identifying vehicles as they drive through checkpoints in cities. However, your samples were collected while vehicles were stationary. As such, this is not a good training set for the classification of moving vehicles, as Doppler shifts and other factors will introduce variations.

-Section 3.3 requires additional detail. None of the SVM hyperparameters are specified.

-The amplitudes in Figure 3 are clearly different. Did you not normalize the data during preprocessing? How did you control for differences in loudness during data collection?

-The vertical axis in Figure 3 does not include units. How did you measure amplitude?

-The sentence on Line 217 ends abruptly.

-In Section 4.3, you manually select discriminative features by looking at plots of the data. This is not a robust practice for machine learning, as specific features may change between datasets. Rather, the employed model should be determining which features to learn from automatically.

-Your training set only consisted of 30 samples, which is far too few for the type of classification you are attempting to perform. It is unlikely this model will be robust to other collected datasets if it learned from such a small group of data collected at only two different distances.

-I don’t understand variant 2 on page 8. Why would the format (mp3 vs wav) matter for classification?

-Your dataset is far too small for k-fold validation. If k=10, you would have only 3 samples in each group.

-Why did you train using only start-up noise in Section 4.4.4? I don’t understand why this is beneficial.

-The label for Figure 13 is mismatched. The percentages along the bottom correspond to the size of the validation set (e.g., 30%), not the training set.

Reviewer 3 Report

In the text, there are long paragraphs with few bibliographic references. The aim of the study needs to be strongly supported in the introduction, and the bibliographic reference per paragraph is only at the end without being able to make a good connection with each statement.

 It is crucial to make it distinct and clear from the bibliographic presentation of the introduction to all the categorical systems available to classify audio identification on the subject. How the article’s writers have chosen which ones exist in the literature and what each one offers perfectly in the related work.

 A study that examined the same topic is listed, but the presentation of these results could be better to help the readers to understand what to expect.

 Polish is also in the text, which must be translated and represented. Also, acronyms such as FFT, SVM, STFT, and others need to be analyzed, and a relevant reference of how they are involved with the analysis of acoustic signals is needed.

 Is there specific literature on the methodology, the quality of the analysis, and the recording in such environments that the study was based on? This is very important to be reported.

Why did the writers choose 44.1 KHz as the primary auditory frequency for analysis? Has the reported freeware application the scientific sensitively and accurately that is needed? What is the best application in other relevant studies on a similar topic? In which conditions was the recording made beyond the distance stated, and what about the environmental noise levels that may affect such sensitive data recordings?

 Good use of MATLAB, but the categorization factors used with which criterion have been selected in the presentation of the results? Authors also list data that fits better with the discussion conclusions, e.g., on lines 197 to 213, and has reference to literature related to their methodology, which should not be there.

 In 217 missing text...

 In 4.4, I am concerned about the five areas of analysis because the authors suggest this way of analysis but need literature to support the theoretical and practical framework of this choice. What about their choice to convert to an MP3 file, it is secure that this does not affect the accuracy of the original recorded message?

 The discussion is feeble, includes information related to the results mainly, and does not have comparative data with existing knowledge or any other relevant research. There are no bibliographic references, the conclusion section is challenging to understand, the main research questions are not clearly answered, and much information is related to the discussion results. Formative changes must be made, and the presentation of the conclusions must be more evident.

Finally, you must follow the instructions of the journal for the bibliography listing.

Reviewer 4 Report

The manuscript entitled "Classification of engine type of vehicle based on audio signal as a source of identification" presents the idea of the classification of various vehicles on the basis of the sound. But the overall the article is not clearly written and missed many crucial details, extensive editing of the overall manuscript is required. Therefore my recommendation is a major revision and rewrite and also put some more experimentation details. 

1) Why authors used different time span for the recording of the engines sounds related to the different car models.

2) For the recording purpose the authors did not clearly explain the specifications and details of the recording device and also many other parameters like SNR values etc are missed.

3) In the abstract important details are also missed, re-write the abstract.

4) In the related work section lines 76-77 should be carefully written better to use word author instead of the full name.

5) In the introduction section the aim, scope, and purpose of the study is also not written , authors must avoid the lengthy paragraphs.

6) In the related work section authors mostly write the full name of the authors frequently, this must be avoided, my suggestion is to re-write this section.

7) The related work section is very limited , better to include the table with the dataset, methodology and the description of the related studies.

8) How the authors measure the effectiveness, is there any formula with reference? 

Round 2

Reviewer 1 Report

The paper is corrected according to the reviewer notes.

Format the paper according to the journal requirements.

Author Response

The paper was corrected accordingly, to suggestions.  We have polished this manuscript by native speaker.

Reviewer 2 Report

I am still a little concerned that you don't know which data are being used to train the network. In other words, is it something distinctive in the petrol/diesel audio waveform or is it something in the background noise that is being used by the network to distinguish the samples. Having so few data makes it difficult to know for certain. This issue arises partly because of the broad variance in Table 5, which includes some classification results as low as 50%. At this point, the algorithm is essentially just guessing randomly. Even if a certain technique did not perform as well as an SVM, a result of 50% leads me to wonder if training is really happening in the way you presume. I think you need to include some type of analysis (e.g., a heat map) to confirm the network is operating in the way you have assumed.

Reviewer 3 Report

Thanks for considering my comments and suggestions and reshaping your work where needed. Everything I suggested has primarily been done, and the results and discussion are now clear and written with better documentation.

Author Response

Thank you for your response.

Reviewer 4 Report

Although the authors improved the manuscript and significant changes have been made, still found a few errors and need improvements. After incorporating these changes the article can be accepted. 

1: At line 361 Fig 47 is mentioned, where is Fig 47?

2:  At line 282 Fig 45 ?

3: In line 330 remove and after table 3.

4: In line 362 the reference number [26] seems to be irrelevant to the context

5: Lines 378-379 look like an incomplete sentence?

6: More then 40-50% references cited by the authors are more then 5 yeras old, and not directly related to the sound classification. My recommendation is to cite few of the latest articles related to sound classification, as shown, 10.3390/sym12111822 , "Environmental sound classification using a regularized DCNN with data augmentation", "Spectral images based environment sound classification using CNN with meaningful data augmentation". etc.

7: My suggestion is to conclude your study in a precise and concise way, alot of irrelevant text must be excluded from the conclusion.

Round 3

Reviewer 2 Report

The authors have adaquately adressed my concerns. I appreciate their detailed response and am now convinced their work is ready for publication.